# Inhibiting *Escherichia coli* Growth by Optimized Low-Power Microwave Irradiation—Delivery of Ag and Au Nanoparticles

**DOI:** 10.3390/molecules30091871

**Published:** 2025-04-22

**Authors:** Yukie Yokota, Nazuna Itabashi, Mari Kawaguchi, Hiroshi Uchida, Nick Serpone, Satoshi Horikoshi

**Affiliations:** 1Department of Materials and Life Sciences, Faculty of Science and Technology, Sophia University, 7-1 Kioicho, Chiyodaku, Tokyo 102-8554, Japan; 2PhotoGreen Laboratory, Dipartimento di Chimica, Università di Pavia, via Taramelli 12, 27100 Pavia, Italy

**Keywords:** low-power microwave-induced transport method, *E. coli* bacteria, Ag and Au nanoparticles, growth inhibition, sterilization

## Abstract

In a ground-breaking recent study, we unveiled the remarkable cellular uptake of 60 nm ZnO and TiO_2_ nanoparticles by NIH/3T3 mouse skin fibroblasts under microwave irradiation. Even more stimulating is our current demonstration of the potent ability of Ag nanoparticles (147 nm) and Au nanoparticles (120 nm) to stifle the growth of *Escherichia coli* (*E. coli*—a prokaryote whose cells lack a membrane-bound nucleus and other membrane-bound organelles), vastly smaller than the NIH/3T3 cells, when exposed to significantly optimized low-power microwave irradiation conditions. Our rigorous assessment of the method’s effectiveness involved scrutinizing the growth rate of *E. coli* bacteria under diverse conditions involving silver and gold nanoparticles. This indisputably underscores the potential of microwave–nanoparticle interactions in impeding bacterial proliferation. Furthermore, our noteworthy findings on the uptake of fluorescent organosilica nanoparticles by *E. coli* cells following brief, repeated microwave irradiation highlight the bacteria’s remarkable ability to assimilate extraneous substances.

## 1. Introduction

In a recent study, we made an unexpected discovery that under specific conditions, nanoparticles have the capacity to infiltrate cells after exposure to low-intensity microwave radiation. This phenomenon was observed when inorganic nanoparticles (NPs) involved in the photocatalysis of TiO_2_ and ZnO, commonly found in sunscreens, were combined with NIH/3T3 mouse skin fibroblast cells and subjected to weak UV light and very mild microwaves, culminating in accelerated cell death (apoptosis) [1]. Initially, it was presumed that this occurrence was solely attributable to the photocatalyst’s interaction with ultraviolet light. However, further investigation revealed that the influence of weak microwaves facilitated the internalization of these NPs into the cells, thereby instigating a photocatalytic reaction within the intracellular environment, ultimately promoting cell death. This revelation underscores the potential hazards associated with everyday products and thus warrants further exploration.

In this regard, a recent articulate review article by Deruelle [2] pointed out that contrary to cosmic microwaves, polarized microwaves used in wireless communication technologies can potentially lead to unbalanced activation of electrosensitive ion channels in cell membranes and thus trigger several biological effects, up to and including DNA damage, cell death, and even cancer, among others (e.g., neurologic, endocrine gland activity, cardiovascular, hemodynamic, metabolic, gastric, ocular, etc.), with exposures below intensities for heating having been known since the 1970s. The 2020 guidelines of the International Commission on Non-ionizing Radiation Protection (ICNIRP) permit an average whole body to be exposed to frequencies between 2 and 300 GHz for a 30 min period to no more than 1000 µW cm^−2^, while for 5G technology (frequencies > 6 GHz) the ICNIRP permitted a local exposure of 20,000 µW cm^−2^ for 6 min over an area of 4 cm^2^; other experts [3], however, suggest a maximum intensity exposure to less than 0.0010 µW cm^−2^. Connected to these concerns and based on limited evidence of increased risk for brain tumors in cell phone users, in 2011 the International Agency for Research on Cancer (IARC-WHO) classified radiofrequency electromagnetic fields (RF-EMFs; frequency range, 30 kHz–300 GHz) as “possibly carcinogenic to humans” (Group 2B) [4]. Since then, further research has established that RF radiation can be classified as a human carcinogen, Group 1 [5].

Bacterial infections have caused significant harm to people, particularly with the emergence of multidrug-resistant bacteria, which has rendered traditional antibiotic therapy relatively ineffective. As a possible remedy for this, hyperthermia resulting from microwave exposure has emerged as a promising treatment for bacterial infections, as microwaves (MWs) are electromagnetic waves in the appropriate frequency that can rapidly heat materials. Mechanistically, an advantage of this latter feature is that both MW thermal and non-thermal effects can lead to the inactivation of various bacteria. In this regard, however, Zhang and coworkers [6] have recently noted that an understanding of microwave radiation in the context of bacteria is not yet mature enough for widespread application. Nonetheless, they did provide some insights into the possible mechanism for bacterial death.

Moreover, recent studies have shown that silver nanoparticles (Ag NPs) can inhibit bacterial proliferation in water treatment [7,8]. Notably, Ag-based antimicrobial nanoparticles have garnered widespread attention due to their remarkable antibacterial efficacy against a diverse spectrum of bacteria [9]. Furthermore, Ag NPs have been identified as primarily inducing oxidative DNA damage [10]. However, the recovery of these nanoparticles after sterilization poses significant challenges due to their size range, which spans a few to several tens of nanometers; introducing precious metal nanoparticles of 100 nm (or larger) into *E. coli* has demonstrated the capacity to diminish the growth rate of *E. coli*, thereby enhancing the sterilization rate and facilitating subsequent recovery. Notably, the uptake of silver nanoparticles (Ag NPs) has been correlated with the induction of cell apoptosis [11].

Related to the present work, Qiao et al. [12] reported that mixing *Garcinia* NPs with Gram-negative bacteria resulted in a temporary antibacterial effect; however, under microwave irradiation, the NPs were taken up by the cells, significantly enhancing the bactericidal effect. In the interim, Bhardwaj et al. [13] reported that the bactericidal rate was considerably enhanced by irradiating *E. coli*, *Staphylococcus aureus*, and *Salmonella typhimurium* with microwaves using Ag nanoparticles. In addition to our studies and those of others, microwave irradiation unmistakably induces the entry of NPs into cells.

Therefore, the primary objective of our current study was to determine the optimal microwave irradiation conditions for suppressing the growth rate of *E. coli* using a low-power microwave method to enhance the bactericidal effect of actively introducing nanoparticles into *E. coli*. The primary challenge was to develop a technique for effectively introducing nanoparticles (NPs) into cells. Successful research in this area can significantly enhance the practical applicability of these techniques. Specifically, the uptake of nanoparticles by *E. coli* cells, which are smaller than NIH/3T3 cells, has the potential to significantly advance the practicality of these studies.

## 2. Results and Discussion

### 2.1. Establishing the Experimental Protocols

Before commencing the main study, initial experiments were conducted to establish specific experimental parameters, focusing on the heating efficacy of the microwave apparatus. The primary goal was to investigate the effect of microwave irradiation on the growth rate of *E. coli* after introducing metal nanoparticles to isolate any potential impact of microwave-induced heating in an untreated aqueous solution. As a result, the power output of the microwave device was consistently set at 10, 20, 30, and 40 W, with each of the four reactor vessels receiving one-quarter of the total power. The temperature fluctuations of the solutions under continual irradiation are graphically represented in Figure 1a.

Temperature variations were evident during microwave exposure at 30 and 40 W power levels. The temperature increment was approximately 1 °C following 10 min of microwave irradiation at 30 W, reaching around 1.4 °C and 2.5 °C after 30 min at 40 W. In contrast, no discernible rise in temperature was observed after 30 min of 20 W microwave irradiation, attributable to the partial dissipation of heat into the surrounding atmosphere. **Experiment III** involved a 7 h microwave irradiation employing 15 W microwaves, deviating from the original plan, which utilized 10 W and 20 W microwaves.

In this investigation, we aimed to assess the potential inhibitory effect of hexadecyl trimethyl ammonium chloride (CTAC) and citric acid, which serve as protective agents for Au and Ag nanoparticles, respectively, on the growth rate of *E. coli*. Equal amounts (100 μL) of these protective agents were introduced to each *E. coli* sample, mirroring the quantities present in the nanoparticle suspensions (Figure 1b). Our observations indicated that, in comparison to the control, no discernible reduction in the growth rate of *E. coli* was evident across any of the tested conditions. Furthermore, upon doubling the volume of the protective agents to 200 μL, no alterations in the growth rate of *E. coli* were discerned. Consequently, the quantity of protective agents applied in this investigation elicited no discernible impact on the growth rate of *E. coli*.

### 2.2. Experiment I

In the case of Ag NPs, the control sample, which consisted of Escherichia coli (*E. coli*) alone, showed an 86-fold increase in growth after a 7 h incubation period. The growth of *E. coli* exhibited a time-dependent increase across all experimental conditions, encompassing the presence of Ag NPs alone, exclusive microwave irradiation (the control), and simultaneous microwave irradiation in the presence of Ag NPs (Ag/MW), with 15 W microwaves uniformly employed. The growth data evince minimal deviation from the control, as illustrated in Figure 2a. Similar to those with silver nanoparticles, comparable trends in E. coli growth were observed in experiments involving gold nanoparticles (Figure 2b). Notably, the presence of noble metal nanoparticles within the *E. coli* bacterial system indicated that, within the scope of experimental error, a 10 min microwave irradiation period minimally impacted *E. coli* growth, if at all. Furthermore, it was ascertained that the mere mixing of Ag or Au NPs was insufficient to impede the proliferation of *E. coli.*

### 2.3. Experiment II

When exposed to microwave irradiation alone or in the presence of either Ag or Au metal nanoparticles alone, *E. coli* exhibited growth over time that did not differ significantly from the control growth, as shown in Figure 3. However, when *E. coli* bacteria were exposed to microwave irradiation in the presence of Ag NPs, a notably decreased growth rate was observed at all times compared to the control, with a 47% reduction after 7 h (Figure 3a). Similarly, exposure to Au NPs and microwave irradiation resulted in a 48% reduction in growth after 7 h relative to the control (Figure 3b). These results indicate that the cellular uptake of noble metal nanoparticles (NPs) by *E. coli* cells led to a significant decrease in growth, suggesting a potential mechanism of cell death through apoptosis.

For the direct assessment of the cellular uptake of nanoparticles by *E. coli* cells in response to microwave exposure, an experimental approach was employed using fluorescent organosilica FITC nanoparticles instead of noble metal NPs. Utilizing fluorescence microscopy, liquid medium samples of *E. coli* bacteria were examined. Notably, the non-microwave-irradiated sample exhibited a distinct separation between *E. coli* cells in the fluid medium and the organosilica FITC NPs at all incubation times (Figure 4a). Furthermore, rapid movement and subsequent disappearance of the organosilica FITC NPs were observed in fluorescent images captured at 30 s intervals. These observations collectively indicate that the nanoparticles exhibited unbound, Brownian-type motion within the liquid medium.

It was also observed that *E. coli* cells exhibited synchronized movement with organosilica FITC NPs when subjected to 10 min intervals of microwave exposure following a 50 min incubation period (Figure 4b). Notably, this behavior contrasted markedly with the movement patterns of individual organosilica FITC NPs. Consistent concurrent drifting of *E. coli* bacteria with internalized organosilica FITC NPs was observed across various incubation times. These observations suggest that microwaves, under specific conditions, facilitate the penetration of approximately 100 nm nanoparticles through the bacterial lipid bilayer membrane, which is predominantly composed of lipopolysaccharides.

In the realm of gold and silver nanoparticles, microwave irradiation has been observed to facilitate the uptake of nanoparticles into *E. coli* cells, resulting in a reduction in growth rate and the demise of certain *E. coli* cells. Two plausible scenarios account for this observation. ***Firstly***, the speculation exists that the nanoparticles may be perceived as foreign entities, thus impeding the biological activity of *E. coli*. ***Secondly***, it is conceivable that the nanoparticles internalized by the cells could function as sensitizers, converting microwave energy into localized heat within the *E. coli* cells, ultimately leading to their demise. Notably, research by Nguyen et al. [14] highlighted the uptake of 15.9 nm Dextran particles, not typically assimilated by cells, following exposure to 18 GHz microwaves. A similar outcome may arise with 2.45 GHz microwaves.

In **Experiment II**, an investigation was conducted to examine the persistence of organosilica FITC nanoparticles within *E. coli* in aqueous environments over 10 days. Notably, the culture temperature was rigorously sustained at 4 °C to inhibit cellular proliferation of *E. coli*. The subsequent fluorescence microscopic assessments, documented in Figure 5, revealed noteworthy outcomes after 3 (Figure 5a) and 10 (Figure 5b) days. Particularly striking was the sustained presence of organosilica FITC nanoparticles within *E. coli* under non-proliferative conditions, substantiating their protracted viability for 3 and 10 days (Figure 5a,b). This highlights the retention of nanoparticles after cellular uptake, indicating their resistance to expulsion following cessation of microwave irradiation.

### 2.4. Experiment III

When subjected to continuous microwave irradiation following the **Experiment III** protocol (see figure of Section 3.4), the growth rate of *E. coli* was hindered by 89%, even in the absence of gold and silver nanoparticles (Figure 6). Moreover, the growth of *E. coli* can be restrained by exposing it to weak microwaves for extended periods without any temperature rise. In the presence of Ag nanoparticles (Ag/MW) and Au nanoparticles (Au/MW), the growth of *E. coli* was further reduced by 27% and 38%, respectively. Interestingly, although we anticipated that Ag/MW would have a lower proliferation rate than Au/MW, Ag exhibited a higher proliferation rate. Additionally, based on the proliferation rate curves in Figure 6, it is evident that the proliferation rate significantly increases after 4 h for Ag NPs/MW and after 5 h for Au NPs/MW. Furthermore, fluorescence microscopy observations confirmed the cellular uptake of organosilica FITC nanoparticles.

The results from fluorescence microscopic observations at 20, 40, and 60 min (Figure 7, inset A) show clear evidence of cellular uptake of the organosilica FITC nanoparticles by *E. coli* bacterial cells following microwave irradiation. However, after 2 and 4 h of continuous irradiation, there was little to no evidence of cellular uptake of the nanoparticles. This suggests that although the nanoparticles may have initially entered the cells during the first 20–60 min of microwave irradiation, they were expelled when the *E. coli* cells were continuously exposed to microwave radiation. Further observations, conducted for 4 h or longer, revealed no nanoparticles within the *E. coli* cells. Thus, we conclude that prolonged continuous microwave irradiation causes the nanoparticles to be expelled and prevents re-entry into the *E. coli* cells under the experimental conditions of **Experiment III**.

### 2.5. Mechanistic Inferences

Although the mechanism of action of microwaves into bactericidal inactivation/death was outside the scope of the present study, some inferences can be made. In this regard, in a comparison of the thermal effects of microwaves on cell membranes, cell walls, soluble chemical oxygen demand, and enzyme inactivation and dysfunction, Cao and coworkers [15] showed that the microwaves’ non-thermal effects (at <40 °C) were more effective in destroying microorganisms, albeit the type of cellular damage caused by each of these two mechanisms seems to remain a matter of controversy. The action of microwaves on bacteria at low power levels and short exposure times at non-lethal temperatures results in a change in the permeability of the bacterial membrane. However, when microwaves act on bacteria at high power levels and prolonged exposure times at lethal temperatures, they cause irreversible damage to the bacterial membrane and cytoplasmic membrane, which can lead directly to bacterial death [6]. Reactive oxygen species also appear to play a non-insignificant role in bactericidal activity, which occurs through a decrease in antioxidant pathways and so contributes to the inactivation of *E. coli* bacteria, for instance [16]. Shaw and coworkers [16] further demonstrated oxidative stress-mediated DNA damage induced by microwaves. For example, exposure of *E. coli* bacteria to microwave radiation decreases the transcriptional gene expression levels of six genes involved in oxidative stress.

Within the present context, after comparing the **Experiment II** and **Experiment III** protocols, we observed that when *E. coli* cells were subjected to incubation followed by 10 min of microwave irradiation, there was a noticeable cellular uptake of nanoparticles. However, when the cells were exposed to continuous microwave irradiation, the rate of cellular uptake of the nanoparticles decreased significantly (see Figure 7). We hypothesize that this can be attributed to the mechanism whereby continued microwave irradiation leads to the expulsion of nanoparticles from the cells, resulting from the relaxation of the cell membrane following cellular uptake. It appears that pulsed microwave irradiation, in brief 10 min periods, plays a crucial role in the cellular uptake of nanoparticles (transport) and possibly other foreign substances by *E. coli* cells.

## 3. Materials and Methods

### 3.1. Escherichia coli (E. coli) Sample

The experiments utilized *Escherichia coli* DH5α competent cells (Takara Bio, Kusatsu, Japan) that had been genetically modified to express an ampicillin-resistant gene. Following the retrieval of *E. coli* colonies from a streak culture, the samples were incubated overnight in 4 mL of Luria–Bertani (LB) medium supplemented with 16 μL of 25 mg/mL ampicillin at 37 °C.

### 3.2. Preparation of Nanoparticles

All reagents were sourced from the Fujifilm Wako Pure Chemical Corporation (Osaka, Japan). The gold and silver nanoparticles were synthesized in aqueous solution using a seed-mediated growth method [17,18]. The silver nitrate solution (50 mL, 2 mM) was prepared using ultrapure water. While stirring, the trisodium citrate dehydrate (0.045 g) was added to the silver nitrate solution. The mixed solution was stirred, and NaBH_4_ solution (1 mL, 0.05 M) was added. After the addition, the silver nanoparticles were generated by irradiation with visible light at room temperature for 15 min (Asahi Spectra Co., Ltd., Tokyo, Japan, MAX-350, 300 W Xe lamp with a band-pass filter at 660 nm), producing a green-colored sol [18]. The concentration of Ag nanoparticles was 0.308 × 10^−8^ M. The seed solution for gold nanoparticles was prepared by adding 9750 μL of ultrapure water, 0.320 g of CTAC, and 250 μL of 0.01 M tetrachloroauric (III) acid trihydrate to a vial. This was followed by adding 450 μL of 0.02 M sodium borohydride as the reducing agent [17]. For the growth solution, 9380 μL of ultrapure water, 0.320 g of CTAC, and 250 μL of 0.01 M tetrachloroauric (III) acid trihydrate were mixed in a vial. Then, 10 μL of a sodium bromide solution was added. The seed and growth solutions were incubated in a water bath at 30 °C for 1 h. Before the reaction commenced, 75 μL of 0.04 M L-ascorbic acid was added to the growth solution as the reducing agent. For the various experiments, 25 μL of the seed solution was transferred to the growth solution and stirred for 5 s. Afterward, the growth solution was left to stand at ambient temperature for one week. The concentration of Ag nanoparticles was 0.702 × 10^−11^ M. CTAC was used as a protective agent for the gold nanoparticles. At the same time, citric acid served a similar purpose for the silver nanoparticles, preventing agglomeration.

The size distribution of noble metal nanoparticles was determined using dynamic light scattering (DLS: Otsuka Electronics Co., Ltd., Osaka, Japan, DLS-8000). The average size of the gold nanoparticles (Au NPs) was 147 nm, while that of the silver nanoparticles (Ag NPs) was 120 nm. The organosilica FITC nanoparticles, with a diameter of 100 nm and emitting fluorescence at 523 nm, were utilized as obtained from Tokyo Chemical Industry Co., Ltd. (Tokyo, Japan).

### 3.3. Microwave Irradiation Equipment

A prototype microwave irradiation apparatus featuring a stainless-steel microwave multi-mode applicator was employed in the experiment. Four patch antennas were situated at the base of the applicator, as depicted in Figure 8. Atop the antennas, a fluororesin board supported four reactors in the form of Pyrex 25 mL triangular flasks. The direct microwave irradiation of the lower section of the reactors was validated, with the reflected microwaves registering at less than 1.0 W. The continuous microwave radiation emanated from a 2.45 GHz microwave semiconductor generator that housed an isolator (LEANFA Srl, Ruvo di Puglia, Italy; model GEAP-A003A01; maximum power, 250 W).

The microwaves produced were apportioned to the four patch antennas using a divider (Mini-circuit ZB4PD-282-50W+, Brooklyn, NY, USA). The semiconductor generator, divider, and patch antenna were interconnected using an N-type cable. An orifice was created on the side of the applicator to accommodate an optical fiber thermometer (FL-2000, Anritsu Meter Co., Ltd., Tokyo, Japan) for monitoring the temperatures of the aqueous solutions within the applicator. In a preliminary trial, an equivalent volume of aqueous solution was placed in four identical reactors and subjected to continuous irradiation with 30-watt microwaves for 30 min. The temperature differential among the aqueous solutions was ±0.9 °C, whereas the microwave electric field strength variance at the four locations was deemed negligible.

### 3.4. Experimental Protocol

The samples of *E. coli* solution in LB medium underwent a 7 h incubation period, during which their spectral absorbance at 600 nm was measured hourly to monitor the growth rate using the turbidity method. The incubation temperature was meticulously maintained at 37 ± 2 °C throughout all experiments, with controls conducted in a shaker or water bath. Subsequently, 100 mL of nanoparticle solution (Ag NP: ca. 4.0 μmol L^−1^, Au NP: ca. 9.3 μmol L^−1^) was introduced to the *E. coli* sample at the onset of the experiment. Three distinct microwave irradiation conditions were implemented to ascertain conditions that would minimize the proliferation of *E. coli*. Schematics of these experimental conditions are presented in Figure 9. Each of the three experiments was conducted three times, with the observed disparities in the experimental data proving statistically insignificant (standard deviation was less than 2%). Nevertheless, the reported data represent the average outcome derived from these trials.

**Experiment I**: Following a 50 min incubation period, the *E. coli* samples underwent irradiation with 15 W (low-power) microwaves for 10 min at a frequency of 2.45 GHz, followed by an additional 6 h incubation period.

**Experiment II**: After a 50 min incubation period, the *E. coli* samples were similarly irradiated with 15 W microwaves for 10 min. This process was repeated hourly for an additional 6 h.

**Experiment III**: The *E. coli* samples were continuously irradiated with 15 W microwaves for 7 h. Owing to the impracticability of culturing with a shaker, the air was continuously bubbled throughout the reaction solution using a silicone tube while the samples were subjected to microwave irradiation.

### 3.5. Analytical Methods

*E. coli*’s growth rate was determined by quantifying the absorbance intensity at 600 nm using a Shimadzu Co. BioSpec Mini Nucleic Acid Protein Spectrophotometer (Kyoto, Japan). Microscopic analysis was performed using a Carl Zeiss AG Axiovert 200 fluorescence microscope (Oberkochen, Germany). *E. coli* samples were positioned on a glass-bottom dish with a cover glass, and FITC fluorescence was observed and captured using a 63× objective lens.

## 4. Concluding Remarks

The cellular uptake of exogenous substances by *E. coli* cells in the presence of microwave irradiation was investigated, utilizing gold and silver nanoparticles and fluorescent organosilica nanoparticles. The experiment involved repeated 10 min microwave irradiation sessions within each hour, complemented by osmotic culture conditions, resulting in a reduction in the growth rate of *E. coli* cells in the presence of noble metal nanoparticles. Although individual silver or gold nanoparticles did not manifest a bactericidal effect, the application of mild microwave pulses induced a bactericidal effect, effectively impeding the proliferation of *E. coli*. Notably, continual microwave irradiation hindered the cellular uptake of nanoparticles by *E. coli*. Experimental evidence illustrated the ability of nanoparticles approximately 100 nm in diameter to traverse cell membranes under microwave irradiation despite their size being approximately one-tenth that of *E. coli* cells.

The present study investigated the introduction of particles through microwaves using the sterilization of *E. coli* cells as a model. This technology may be the basis for various applied technologies, such as microwave transfection to achieve intracellular uptake of foreign substances. Transfection (gene introduction) is a method by which foreign nucleic acids can be inserted into eukaryotic cells. There are various methods for transfection, including chemical and physical methods, each of which has advantages and disadvantages [19]. Our goal was that if transfection could be accomplished by irradiating with microwaves, it might be possible for transfection to occur in a milder environment.

## Figures and Tables

**Figure 1 molecules-30-01871-f001:**
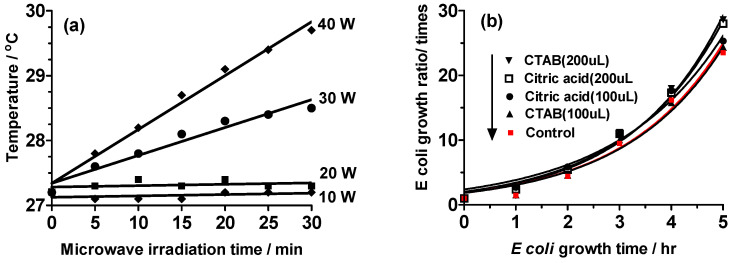
(**a**) Temperature changes in the aqueous solutions at each microwave output power (10, 20, 30, and 40 W); (**b**) changes in the growth ratio of *E*. *coli* caused by the addition of protective agents under irradiation with 20 W microwaves.

**Figure 2 molecules-30-01871-f002:**
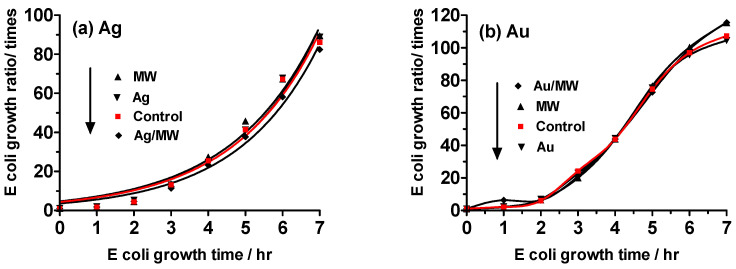
Growth ratios of *E*. *coli* against incubation time monitored by a turbidity method. (**a**) Ag NPs and (**b**) Au NPs under **Experiment I** conditions with 15-watt microwave irradiation (MW) for 10 min after 50 min incubation followed by an additional 6 h continuous incubation period (see see figure of Section 3.4).

**Figure 3 molecules-30-01871-f003:**
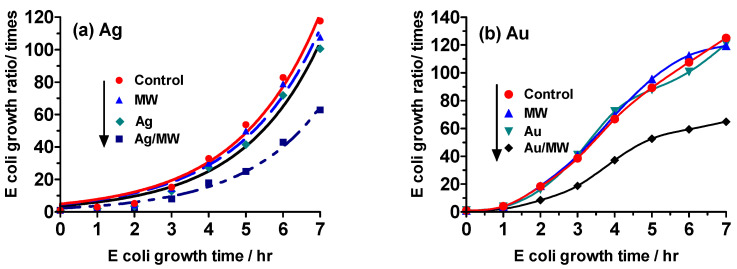
Growth ratios of *E*. *coli* using the turbidity method under **Experiment II** conditions in which the *E. coli* system was microwave-irradiated (MW) for 10 min after a 50 min incubation period, with the process repeated every hour for an additional 6 hr period. (**a**) Ag NPs; (**b**) Au NPs.

**Figure 4 molecules-30-01871-f004:**
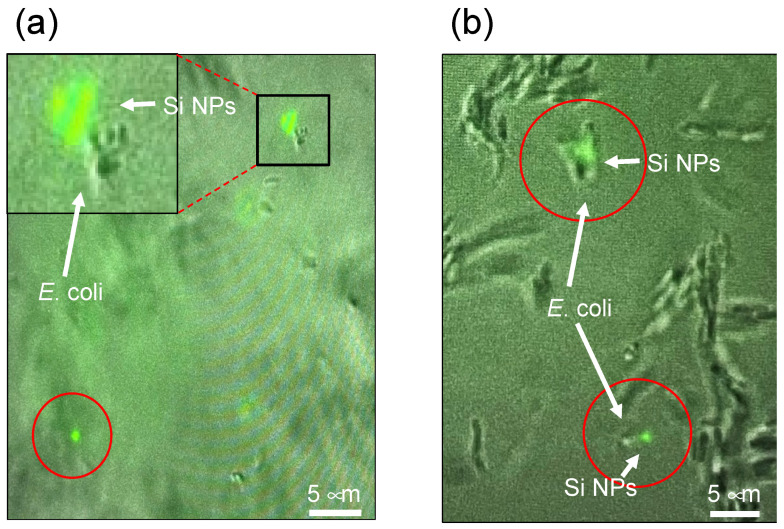
Fluorescence microscopic images for *E*. *coli* under “**Experiment II**” protocol (see figure of Section 3.4) in the presence of fluorescent organosilica FITC NPs (Si NPs) in which microwave irradiation was repeated 10 min after successive 50 min incubation periods: (**a**) no microwave irradiation, (**b**) under microwave irradiation.

**Figure 5 molecules-30-01871-f005:**
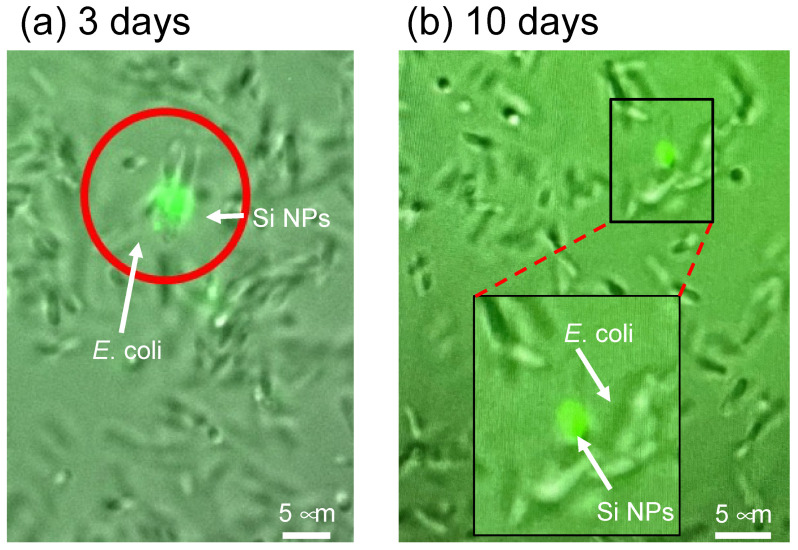
Fluorescence microscopic images of E. coli and conditions with organosilica FITC NPs (Si NPs) after standing still for (**a**) 3 days and (**b**) 10 days at a temperature of 4 °C under the conditions of **Experiment II**.

**Figure 6 molecules-30-01871-f006:**
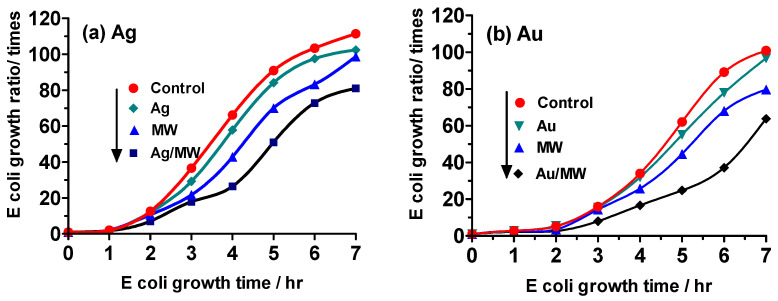
Growth ratios of *E*. *coli* using the turbidity method under the **Experiment III** protocol that implicated continuous microwave irradiation (MW) with concomitant bubbling air into the reactor vessels: (**a**) Ag NPs, (**b**) Au NPs.

**Figure 7 molecules-30-01871-f007:**
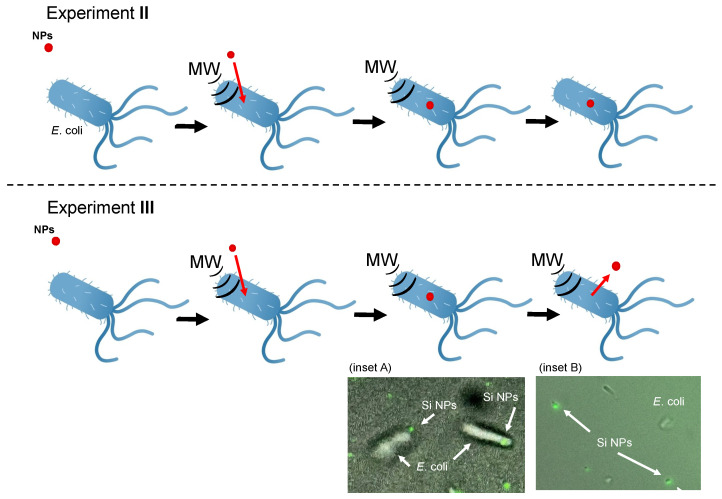
Schematic image of nanoparticle uptake into *E*. *coli* under **Experiment II** and **Experiment III** conditions. Inset A: fluorescence microscopic image of organosilica FITC NPs (Si NPs) taken up into *E. coli* after 20 min of microwave irradiation under conditions of **Experiment II**. Inset B: fluorescence microscopic image of organosilica FITC NPs exiting the *E*. *coli* cell after 2 hr of microwave irradiation under conditions of **Experiment III**.

**Figure 8 molecules-30-01871-f008:**
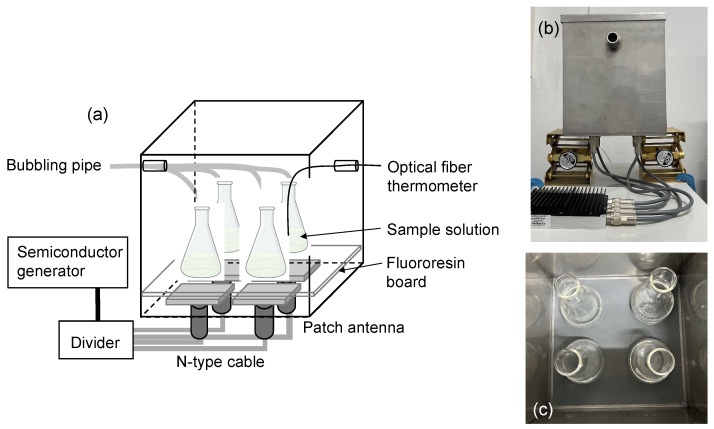
(**a**) Schematic image of the microwave irradiation equipment, consisting of a 2.45 GHz semiconductor microwave generator, an optical fiber thermometer, a divider, four patch antennas, and a multi-mode applicator; (**b**) photograph of the multi-mode applicator and divider; (**c**) photograph displaying the inner space of the multi-mode applicator.

**Figure 9 molecules-30-01871-f009:**
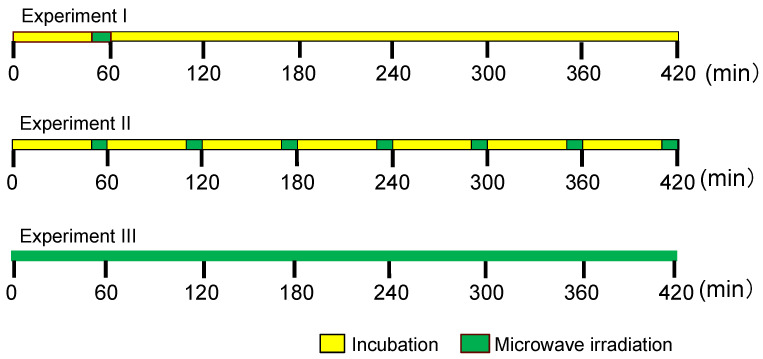
Schematic of the timetable of the relationship between microwave irradiation and culture incubation times in **Experiments I, II,** and **III**.

## Data Availability

Data are contained within the article.

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
