# Peer review of "Inhibiting Escherichia coli Growth by Optimized Low-Power Microwave Irradiation—Delivery of Ag and Au Nanoparticles"

_molecules, 2025, doi:10.3390/molecules30091871_

Round 1
Reviewer 1 Report
Comments and Suggestions for Authors
Dear authors, while reviewing the article, the following questions arose:
- It is necessary to significantly expand the literature review on the research topic, and references to sources should not exceed 5 years
- What is the scope of application of the obtained research results
- In the Russian-language source [https://cyberleninka.ru/article/n/vliyanie-mikrovolnovogo-izlucheniya-razlichnyh-chastot-na-rost-kultur-escherichia-coli/viewer], in the conclusions (item 3), information is provided on the stimulation of the growth of E. coli at a microwave radiation frequency of 2 GHz. For what reasons was the radiation frequency of 2.45 GHz adopted, were other frequency ranges considered?
- What strain of E. coli was used in the studies?
- How were the bacteria counted? Specify the unit of measurement
- How long does the effect of inhibiting the growth of E. coli last by introducing colloidal silver and gold particles?
Author Response
- It is necessary to significantly expand the literature review on the research topic, and references to sources should not exceed 5 years.
We have updated some of the references. However, this study is rare, and there are not many references. Therefore, some of the literature is more than 5 years old. The latest ones cannot replace these old references, and these papers are what readers need.
- What is the scope of application of the obtained research results?
The second paragraph now completes this. In the third paragraph, we connect microwaves to bacterial infections.
- In the Russian-language source [https://cyberleninka.ru/article/n/vliyanie-mikrovolnovogo-izlucheniya-razlichnyh-chastot-na-rost-kultur-escherichia-coli/viewer], in the conclusions (item 3), information is provided on the stimulation of the growth of E. coli at a microwave radiation frequency of 2 GHz. For what reasons was the radiation frequency of 2.45 GHz adopted, were other frequency ranges considered?
For the final application of this research, it will be more practical if the microwave frequency is in the ISM band (Industrial Scientific and Medical Band). We used 2.45 GHz, which is the most common frequency worldwide.
- What strain of E. coli was used in the studies?
The type of Escherichia coli and where to obtain it are specified at the beginning of Section 3.1.
- How were the bacteria counted? Specify the unit of measurement.
We did not count E. coli. We measured growth rates as reported in the paper.
- How long does the effect of inhibiting the growth of E. coli last by introducing colloidal silver and gold particles?
Thank you for your question. We have not performed experiments longer than 7 hours. This paper discusses the phenomenon; we will discuss this point further in development and application research.
Reviewer 2 Report
Comments and Suggestions for Authors
The authors have made useful explorations in the combination of nanoparticles and microwave radiation for sterilization, and have achieved remarkable research results. However, the article still could to be enhanced with the following:
- In the title, "E coli" should be replaced by "Escherichia coli" to maintain the integrity of the scientific terminology
- "Introduction" section, first paragraph. The authors mention the effects of radiation on cells, such as apoptosis, and the significance and mechanisms of this need to be further explored, provide a more detailed description of the significance of the study and possible mechanisms, such as how microwave radiation causes cell death (Reference: PMID: 39640342 and PMID: 39939798).
- In “Materials and methods” (e.g., specific concentrations of nanoparticles, specific temperature control of experiments, etc.), more detailed parameter information can be provided to ensure the reproducibility of the experiment.
- In” Results and Discussion”, It is necessary to discuss how the results of this study can be applied to sterilization or to study the uptake of exogenous substances or apoptosis in cells, as well as the possible reasons for different results, such as the application of mild microwave to induce bactericidal death and effectively prevent the proliferation of E. coli. Notably, continuous microwave radiation hindered cellular uptake of nanoparticles by E. coli.
5.Line 13, "protist" is used in the original sentence "protist vastly smaller than the NIH/3T3 cells", but E. coli is a prokaryotic and should be changed to "prokaryote".
- Line 172, E.coli should be “E. coli”.
- Line 314, E.coli should be “E. coli”.
Check and fix any grammatical or spelling mistakes.
Author Response
Quality of English Language
(x) The English could be improved to more clearly express the research.
( ) The English is fine and does not require any improvement.
We have reviewed the entire manuscript and made necessary changes to the text to address the issues raised by this reviewer.
Comments and Suggestions for Authors
The authors have made useful explorations in the combination of nanoparticles and microwave radiation for sterilization and have achieved remarkable research results. However, the article still could be enhanced with the following:
We thank the reviewer for the constructive critiques.
- In the title, "E coli" should be replaced by "Escherichia coli" to maintain the integrity of the scientific terminology
DONE. Thank you
- "Introduction" section, first paragraph. The authors mention the effects of radiation on cells, such as apoptosis, and the significance and mechanisms of this need to be further explored, provide a more detailed description of the significance of the study and possible mechanisms, such as how microwave radiation causes cell death (Reference: PMID: 39640342 and PMID: 39939798).
We have added a paragraph (the second) to address this issue, and we thank the reviewer for pointing out the two references we consulted, together with two additional references relevant to the topic. Furthermore, we have added Section 2.4 regarding mechanistic inferences and two additional references.
- In “Materials and methods” (e.g., specific concentrations of nanoparticles, specific temperature control of experiments, etc.), more detailed parameter information can be provided to ensure the reproducibility of the experiment.
More detailed information on the synthesis of nanoparticles has been added to Section 3.2.
- In” Results and Discussion”, It is necessary to discuss how the results of this study can be applied to sterilization or to study the uptake of exogenous substances or apoptosis in cells, as well as the possible reasons for different results, such as the application of mild microwave to induce bactericidal death and effectively prevent the proliferation of E. coli. Notably, continuous microwave radiation hindered the cellular uptake of nanoparticles by E. coli.
Thank you for your valuable comments. The conclusion section shows the reviewer’s examples, to which we added new potential applications.
- Line 13, "protist" is used in the original sentence "protist vastly smaller than the NIH/3T3 cells", but coli is a prokaryotic and should be changed to "prokaryote".
Changed. Thank you
- Line 172, E.coli should be “E. coli”. DONE. Thank you
- Line 314, E.coli should be “E. coli”. DONE. Thank you
Comments on the Quality of English Language: Check and fix any grammatical or spelling mistakes.
DONE
Round 2
Reviewer 1 Report
Comments and Suggestions for Authors The authors have provided full responses to the comments.I believe that the article can be accepted in its current form.